# Finite temperature quantum annealing solving exponentially small gap problem with non-monotonic success probability

Anurag Mishra [1,2], Tameem Albash[1,2,3] & Daniel A. Lidar[1,2,4,5]

Closed-system quantum annealing is expected to sometimes fail spectacularly in solving simple problems for which the gap becomes exponentially small in the problem size. Much less is known about whether this gap scaling also impedes open-system quantum annealing. Here, we study the performance of a quantum annealing processor in solving such a problem: a ferromagnetic chain with sectors of alternating coupling strength that is classically trivial but exhibits an exponentially decreasing gap in the sector size. The gap is several orders of magnitude smaller than the device temperature. Contrary to the closed-system expectation, the success probability rises for sufficiently large sector sizes. The success probability is strongly correlated with the number of thermally accessible excited states at the critical point. We demonstrate that this behavior is consistent with a quantum open-system description that is unrelated to thermal relaxation, and is instead dominated by the system's properties at the critical point.

[1] Department of Physics and Astronomy, University of Southern California, Los Angeles, CA 90089, USA. [2] Center for Quantum Information Science & Technology, University of Southern California, Los Angeles, CA 90089, USA. [3] Information Sciences Institute, University of Southern California, Marina del Rey, CA 90292, USA. [4] Department of Electrical Engineering, University of Southern California, Los Angeles, CA 90089, USA. [5] Department of Chemistry, University of Southern California, Los Angeles, CA 90089, USA. Correspondence and requests for materials should be addressed to A.M. (email: anuragmi@usc.edu)

Quantum annealing (QA)[1–8], also known as the quantum adiabatic algorithm[9,10] or adiabatic quantum optimization[11,12] is a heuristic quantum algorithm for solving combinatorial optimization problems. Starting from the ground state of the initial Hamiltonian, typically a transverse field, the algorithm relies on continuously deforming the Hamiltonian such that the system reaches the final ground state—typically of a longitudinal Ising model—thus solving the optimization problem. In the closed-system setting, the adiabatic theorem of quantum mechanics[13] provides a guarantee that QA will find the final ground state if the run-time is sufficiently large relative to the inverse of the quantum ground state energy gap[14,15]. However, this does not guarantee that QA will generally perform better than classical optimization algorithms. In fact, it is well-known that QA, implemented as a transverse field Ising model, can result in dramatic slowdowns relative to classical algorithms even for very simple optimization problems[12,16–19]. Generally, this is attributed to the appearance of exponentially small gaps in such problems[20].

A case in point is the ferromagnetic Ising spin chain with alternating coupling strength and open boundary conditions studied by Reichardt[12]. The "alternating sectors chain" (ASC) of length $N$ spins is divided into equally sized sectors of size $n$ of "heavy" couplings $W_1$ and "light" couplings $W_2$, with $W_1 > W_2 > 0$. Since all the couplings are ferromagnetic, the problem is trivial to solve by inspection: the two degenerate ground states are the fully aligned states, with all spins pointing either up or down. However, this simple problem poses a challenge for closed-system QA since the transverse field Ising model exhibits an exponentially small gap in the sector size $n$[12], thus forcing the run-time to be exponentially long in order to guarantee a constant success probability. A related result is that QA performs exponentially worse than its imaginary-time counterpart for disordered transverse field Ising chains with open boundary conditions[21], where QA exhibits an infinite-randomness critical point[22].

As a corollary, we may naively expect that for a fixed run-time, the success probability will decrease exponentially and monotonically with the sector size. While such a conclusion does not follow logically from the adiabatic theorem, it is supported by the well-studied Landau–Zener two-level problem[23–25]. How relevant are such dire closed-system expectations for real-world devices?

By varying the sector size of the ASC problem on a physical quantum annealer, we find a drastic departure from the above expectations. Instead of a monotonically decreasing success probability (at constant run-time), we observe that the success probability starts to grow above a critical sector size $n^*$, which depends mildly on the chain parameters ($W_1$, $W_2$). We explain this behavior in terms of a simple open-system model whose salient feature is the number of thermally accessible states from the instantaneous ground state at the quantum critical point. The scaling of this "thermal density of states" is nonmonotonic with the sector size and peaks at $n^*$, thus strongly correlating with the success probability of the quantum annealer. Our model then explains the success probability behavior as arising predominantly from the number of thermally accessible excitations from the ground state, and we support this model by adiabatic master equation simulations. Our result does not imply that open-system effects can lend an advantage to QA, and hence it is different from proposed mechanisms for how open-system effects can assist QA. For example, thermal relaxation is known to provide one form of assistance to QA[26–29], but our model does not use thermal relaxation to increase the success probability above $n^*$. We note that Ref.[28] introduced the idea that significant mixing due to open-system effects (beyond relaxation) at an anticrossing between the first excited and ground states could provide an advantage, and its theoretical predictions were

supported by the experiments in Ref.[30]. In Ref.[28], an analysis of adiabatic Grover search was performed (a model which cannot be experimentally implemented in a transverse field Ising model), along with numerical simulations of random field Ising models. In contrast, here we treat an analytically solvable model that is also experimentally implementable using current quantum annealing hardware. We also compare our empirical results to the predictions of the classical spin-vector Monte Carlo (SVMC) model[31], and find that it does not adequately explain them. Our study lends credence to the notion that the performance of real-world QA devices can differ substantially from the scaling of the quantum gap.

## Result

**The alternating sectors chain model.** We consider the transverse Ising model with a time-dependent Hamiltonian of the form:

$$H(s) = -A(s) \sum_{i=1}^{N} \sigma_i^x + B(s) H_{\text{ASC}}, \quad (1)$$

where $t_f$ is the total annealing time, $s = t/t_f \in [0, 1]$, and $A(s)$ and $B(s)$ are the annealing schedules, monotonically decreasing and increasing, respectively, satisfying $B(0) = 0$ and $A(1) = 0$. The alternating sectors chain Hamiltonian is

$$H_{\text{ASC}} = -\sum_{i=1}^{N-1} J_i \sigma_i^z \sigma_{i+1}^z, \quad (2)$$

where for a given sector size $n$ the couplings are given by

$$J_i = \begin{cases} W_1 & \text{if } \lceil i/n \rceil \text{ is odd} \\ W_2 & \text{otherwise} \end{cases} \quad (3)$$

Thus the $b+1$ odd-numbered sectors are "heavy" ($J_i = W_1$), and the $b$ even-numbered sectors are "light" ($J_i = W_2$) for a total of $2b + 1 = \frac{N-1}{n}$ sectors. This is illustrated in Fig. 1.

We briefly summarize the intuitive argument of Ref.[12] for the failure of QA to efficiently solve the ASC problem. Consider the $N \gg 1$ and $n \gg 1$ limit, where any given light or heavy sector resembles a uniform transverse field Ising chain. Each such transverse field Ising chain encounters a quantum phase transition separating the disordered phase and the ordered phase when the strength of the transverse field and the chain coupling are equal, i.e., when $A(s) = B(s)J_i$[32]. Therefore, the heavy sectors order independently before the light sectors during the anneal. Since the transverse field generates only local spin flips, QA is likely to get stuck in a local minimum with domain walls (antiparallel spins resulting in unsatisfied couplings) in the disordered (light) sectors, if $t_f$ is less than exponential in $n$. We note that this mechanism, in which large local regions order before the whole is well-known in disordered, geometrically local optimization problems, giving rise to a Griffiths phase[22].

This argument explains the behavior of a closed-system quantum annealer operating in the adiabatic limit. To check its experimental relevance, we next present the results of tests performed with a physical quantum annealer operating at nonzero temperature.

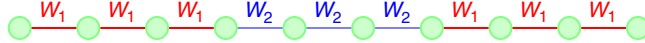

**Fig. 1** Illustration of an alternating sector chain (ASC). This example has sector size $n = 3$, length $N = 10$, and number of sectors $2b + 1 = 3$. Red lines denote the heavy sector with coupling $W_1$, blue lines denote the light sector with coupling $W_2 < W_1$

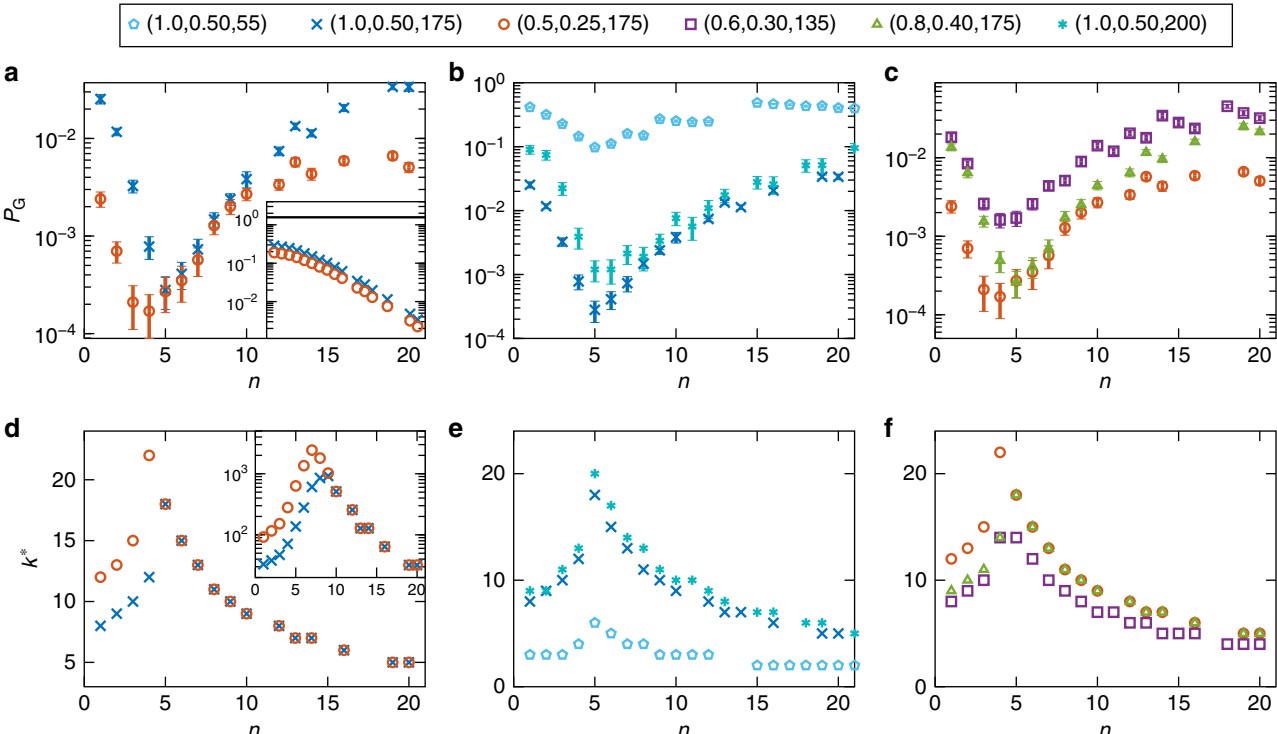

**Fig. 2** Empirical success probability vs. $k^*$ for the ASC problem on the DW2X processor. $k^*$ denotes the number of single-fermion energies that fall below the thermal energy gap at the point of the minimum gap $s^*$. The legend entries indicate the chain parameters: $(W_1, W_2, N)$. The error bars everywhere indicate 95% confidence intervals calculated using a bootstrap over different gauges and embeddings. **a–c** Contrary to closed-system theory expectations, the success probability $P_G$ is nonmonotonic in the sector size $n$, first decreasing and then increasing, exponentially. Inset (**a**): the minimum gap (in GHz) of the chains as a function of the sector size $n \in \{1, \ldots, 20\}$. The solid black line denotes the operating temperature energy scale of the DW2X. **d–f** For all chains we studied the ground state success probability has a minimum at the sector size $n^*$, where the peak in the number of single-fermion states $k^*$ occurs [compare with (**a–c**)]. The rise and fall pattern, as well as the location of $n^*$, are in agreement with the behavior of $P_G$ within the error bars. Inset (**d**): the total number of energy eigenstates that fall below the thermal energy gap as a function of the sector size $n$. In this case the peak position does not agree with the ground state success probability minimum

**Empirical results.** As an instantiation of a physical quantum annealer we used a D-Wave 2X (DW2X) processor. We consider ASCs with sector size $n \in [2,20]$. Since the number of sectors $b = (N-1)/n$ must be an integer, the chain length varies slightly with $n$. The minimum gap for these chains is below the processor temperature. Additional details about the processor and of our implementation of these chains are given in Section Methods.

Figure 2a–c shows the empirical success probability results for a fixed annealing time $t_f = 5\,\mu s$. Longer annealing times do not change the qualitative behavior of the results, but do lead to changes in the success probability (we provide these results in Supplementary Note 7). A longer annealing time can result in more thermal excitations near the minimum gap, but it may also allow more time for ground state repopulation after the minimum gap. The latter can be characterized in terms of a recombination of fermionic excitations by a quantum-diffusion mediated process[33]. Unfortunately, we cannot distinguish between these two effects, as we only have access to their combined effect in the final-time success probability.

In stark contrast to the theoretical closed-system expectation, the success probability does not decrease monotonically with sector size, but exhibits a minimum, after which it grows back to close to its initial value. The decline as well as the initial rise are exponential in $n$. Longer chains result in a lower $P_G$ and a more pronounced minimum, but the position of the minimum depends only weakly on the chain parameter values $(W_1, W_2)$ (the value of $n^*$ shifts to the right as $(W_1, W_2)$ are increased) but not on $N$.

What might explain this behavior? Clearly, a purely gap-based approach cannot suffice, since the gap shrinks exponentially in $n$ for the ASC problem[12] [see also the inset of Fig. 2a]. However, for all chain parameters we have studied, the temperature is greater than the quantum minimum gap. In this setting not only the gap matters, but also the number of accessible energy levels that fall within the energy scale set by the temperature. In an open-system description of quantum annealing[26,34–38], both the Boltzmann factor $\exp(-\beta\Delta)$ ($\beta$ denotes the inverse temperature and $\Delta$ is the minimum gap) and the density of states determine the excitation and relaxation rates out of and back to the ground state. As we demonstrate next, the features of the DW2X success probability results, specifically the exponential fall and rise with $n$, and the position of the minimum, can be explained in terms of the number of single-fermion states that lie within the temperature energy scale at the critical point.

**Fermionization.** We can determine the spectrum of the quantum Hamiltonian [Eq. (1)] by transforming the system into a system of free fermions with fermionic raising and lowering operators $\eta_k^\dagger$ and $\eta_k$[32,39]. The result is[12]:

$$H(s) = E_g(s) + \sum_{k=1}^{N} \lambda_k(s)\eta_k^\dagger \eta_k, \qquad (4)$$

where $E_g(s)$ is the instantaneous ground state energy and $\{\lambda_k(s)\}$ are the single-fermion state energies, i.e., the eigenvalues of the

linear system

$$\Phi_k(s)(A - B)(A + B) = \lambda_k^2(s)\Phi_k(s), \tag{5}$$

where the matrices $A$ and $B$ are tridiagonal and are given in Supplementary Note 1 along with full details of the derivation. The vacuum of the fermionic system $|0\rangle$ is defined by $\eta_k|0\rangle = 0 \forall k$ and is the ground state of the system. Higher energy states correspond to single and many-particle fermionic excitations of the vacuum. At the end of the anneal, fermionic excitations corresponds to domain walls in the classical Ising chain (see Supplementary Note 2).

The Ising problem is $\mathbb{Z}_2$-symmetric, so the ground state and the first excited state of the quantum Hamiltonian merge toward the end of evolution to form a doubly degenerate ground state. Since any population in the instantaneous first excited state will merge back with the ground state at the end of the evolution, the relevant minimum gap of the problem is the gap between the ground state and the second excited state: $\Delta(s) = \lambda_2(s)$, which occurs at the point $s^* = \mathrm{argmin}_{s \in [0,1]} \Delta(s)$. In the thermodynamic limit, this point coincides with the quantum critical point, where the geometric mean of the Ising fields balances the transverse field, $A(s^*) = \sqrt{W_1 W_2} B(s^*)$[40,41]. Henceforth, we write $\Delta \equiv \Delta(s^*)$ for the minimum gap.

**Spectral analysis.** Let $k^*$ be the number of single-fermion states with energy smaller than the thermal gap at the critical point, i.e.,

$$k^* = \mathrm{argmax}_k\{\lambda_k(s^*) < T\}. \tag{6}$$

As can be seen by comparing Fig. 2d–f to Fig. 2a–c, we find that the behavior of $k^*$ correlates strongly with the ground state success probability for all ASC cases we tested, when we set $T = 12\,\text{mK} = 1.57\,\text{GHz}$, the operating temperature of the DW2X processor (we use $k_B = \hbar = 1$ units throughout). Specifically, $k^*$ peaks exactly where the success probability is minimized, which strongly suggests that $k^*$ is the relevant quantity explaining the empirically observed quantum annealing success probability. Longer chains result in a larger value of $k^*$ and a more pronounced maximum. Of all the ASC sets we tried, we only found a partial exception to this rule for the case (1, 0.5, and 200), where $k^*$ peaks at $n^* = 5$ [Fig. 2e], but the empirical success probability for $n = 5$ and $n = 6$ is roughly the same [Fig. 2b]. We show later that this exception can be resolved when the details of the energy spectrum are taken into account via numerical simulations.

In contrast, the total number of energy eigenstates (including multifermion states) that lie within the thermal gap $[E_g(s^*), E_g(s^*) + T]$, while rising and falling exponentially in $n$ like the empirical success probability in Fig. 2a, does not peak in agreement with the peak position of the latter [see the inset of Fig. 2d].

Why and how does the behavior of $k^*$ explain the value of $n^*$? Heuristically, we expect the success probability to behave as

$$P_G \sim \frac{1 - e^{-\beta\Delta}}{d}, \tag{7}$$

where $d$ is the "thermal density of states" at the critical point $s^*$. Note that the role of the gap here is different from the closed-system case, since we are assuming that thermal transitions dominate over diabatic ones, so that the gap is compared to the temperature rather than the annealing time. Contrast this with the closed-system case, where the Landau–Zener formula for closed two-level systems and Hamiltonians analytic in the time parameter (subject to a variety of additional technical conditions)

states that: $P_G \sim 1 - e^{-\eta\Delta^2 t_f}$, where $\eta$ is a constant with units of time that depends on the parameters that quantify the behavior at the avoided crossing (appearing in, e.g., the proof of Theorem 2.1 in Ref.[25]). Since then $P_G = O(\eta\Delta^2 t_f)$, we expect the success probability to decrease exponentially at constant run-time $t_f$ if the gap shrinks exponentially in the system size.

Our key assumption is that the thermal transitions between states differing by more than one fermion are negligible. That is, thermal excitation (relaxation) only happens via creation (annihilation) of one fermion at a time (see Supplementary Note 3 for a detailed argument). Additionally, the Boltzmann factor suppresses excitations that require energy exchange greater than $\lambda_{k^*}$. Starting from the ground state, all single-fermion states with energy $\leq E_g + \lambda_{k^*}$ are populated first, followed by all two-fermion states with total energy $\leq E_g + \lambda_{k^*} + \lambda_{k^*-1}$, etc. In all, $\sum_{k=1}^{k^*} \binom{k^*}{k} = 2^{k^*} - 1$ excited states are thermally populated in this manner. Thus $d \sim 2^{k^*}$ states are thermally accessible from the ground state.

For a sufficiently small gap we have $1 - e^{-\beta\Delta} \sim \beta\Delta$, so that $P_G \sim \beta\Delta/d$. As can be seen from Fig. 2d–f, $k^*$ rises and falls steeply for $n < n^*$ and $n > n^*$, respectively. For the ASCs under consideration, $d$ varies much faster with $n$ than the gap $\Delta$ (see Fig. 3). Thus $P_G \sim 2^{-k^*}$. This argument explains both the observed minimum of $P_G$ at $n^*$ and the exponential drop and rise of $P_G$ with $n$, in terms of the thermal density of states. In Supplementary Note 4 we give a more detailed argument based on transition rates obtained from the adiabatic master equation, which we discuss next.

**Master equation model.** We now consider a simplified model of the open-system dynamics in order to make numerical predictions. We take the evolution of the populations $\mathbf{p} = \{p_a\}$ in the instantaneous energy eigenbasis of the system to be described by a Pauli master equation[42]. The form of the Pauli master equation is identical to that of the adiabatic Markovian quantum master equation[35], derived for a system of qubits weakly coupled to independent identical bosonic baths. The master equation with an Ohmic bosonic bath has been successfully applied to qualitatively (and sometimes quantitatively) reproduce empirical D-Wave data[43–46]. However, it does not account for $1/f$ noise[47], which

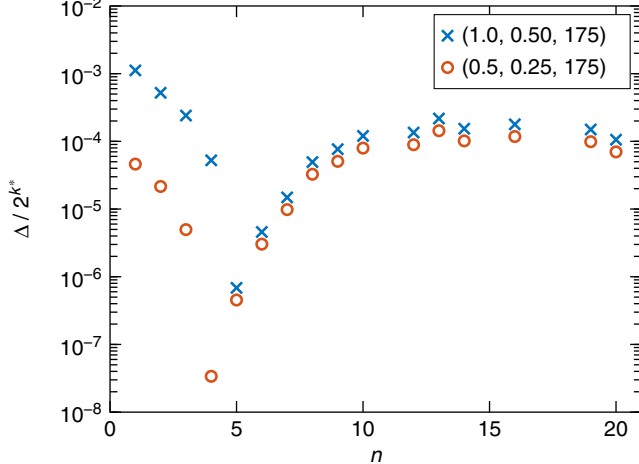

**Fig. 3** Ratio of the gap to the thermal density of states, as a function of sector size. Two alternating sector chain cases are shown. The position of the minimum is determined by $d$ rather than $\Delta$, as can be seen by comparing to Fig. 2d, where the plot of $d = 2^{k^*}$ alone correlates well with the position of minima in the empirical success probability curves

may invalidate the weak coupling approximation when the energy gap is smaller than the temperature[48].

After taking diagonal matrix elements and restricting just to the dissipative (non-Hermitian) part one obtains the Pauli master equation[42] describing the evolution of the population $\mathbf{p} = \{p_a\}$ in the instantaneous energy eigenbasis of the system:[38]

$$\frac{\partial p_a}{\partial t} = \sum_{b \neq a} \gamma(\omega_{ba}) M_{ab} p_b - \sum_{b \neq a} \gamma(\omega_{ab}) M_{ba} p_a. \quad (8)$$

Here all quantities are time-dependent and the matrix elements are

$$M_{ab}(s) = \sum_{\alpha=1}^{N} |\langle a(s)|\sigma_\alpha^z|b(s)\rangle|^2, \quad (9)$$

where we have assumed an independent thermal bath for each qubit $\alpha$ and where the indices $a$ and $b$ run over the instantaneous energy eigenstates of the system Hamiltonian [Eq. 4] in the fermionic representation [i.e., $H(s)|a(s)\rangle = E_a(s)|a(s)\rangle$] and $\omega_{ab} = E_a - E_b$ is the corresponding instantaneous Bohr frequency. Since the basis we have written this equation in is time-dependent, there are additional terms associated with the changing basis[35], but we ignore these terms here since we are assuming that the system is dominated by the dissipative dynamics associated with its interaction with its thermal environment.

The rates $\gamma(\omega)$ satisfy the quantum detailed balance condition[49,50], $\gamma(-\omega) = e^{-\beta\omega}\gamma(\omega)$, where $\omega \geq 0$. In our model each qubit is coupled to an independent pure-dephasing bath with an Ohmic power spectrum:

$$\gamma(\omega) = 2\pi\eta g^2 \frac{\omega e^{-|\omega|/\omega_c}}{1 - e^{-\beta\omega}}, \quad (10)$$

with UV cutoff $\omega_c = 8\pi$ GHz and the dimensionless coupling constant $\eta g^2 = 1.2 \times 10^{-4}$. The choice for the UV cutoff satisfies the assumptions made in the derivation of the master equation in the Lindblad form[35]. Note that we do not adjust any of the master equation parameter values, which are taken from Ref.[43]. Details

about the numerical solution procedure are given in Section Methods, and in Supplementary Note 5 we also confirm that the validity conditions for the derivation of the master equation are satisfied for a relevant range of $n$ values given the parameters of our empirical tests.

Numerically solving the master equation while accounting for all thermally populated $2^{k^*}$ states is computationally prohibitive, but we can partly verify our interpretation by restricting the evolution of the system described in Eq. (8) to the vacuum and single-fermion states. This is justified in Supplementary Note 3, where we show that transitions between states differing by more than a single fermion are negligible. In other words, the dominant thermal transitions occur from the vacuum to the single-fermion states, from the single-fermion states to the two-fermion states, etc. The restriction to the vacuum and single-fermion states further simplifies the master Eq. (8) to:

$$\dot{p}_0 = \sum_b \gamma(\lambda_b) M_b p_b - p_0 \sum_b \gamma(-\lambda_b) M_b \quad (11)$$

$$\dot{p}_i = \gamma(-\lambda_i) M_i p_0 - \gamma(\lambda_i) M_i p_i, \quad (12)$$

where $\{p_b\}_{b=1}^N$ are the single-particle fermion energy populations and $\{\lambda_b\}$ their energies found by solving Eq. (5), and $M_{ab}$ [Eq. (9)] $M_b = \sum_{\alpha=1}^{N} |\langle 0|\sigma_\alpha^z|b\rangle|^2$. For a better approximation that accounts for more states, we can also perform a two-fermion calculation where we keep the vacuum, the first $k^*$ one-fermion states and the next $k^*(k^*-1)/2$ two-fermion states. For two-fermion simulations the master equation becomes Eqs. (11) and (12) along with

$$\dot{p}_i = \gamma(-\lambda_i) M_i p_0 - \gamma(\lambda_i) M_i p_i$$
$$+ \sum_{j \neq i} \gamma(\lambda_j) M_j p_{ij} - p_i \sum_{j \neq i} \gamma(-\lambda_i) M_j \quad (13)$$

$$\dot{p}_{ij} = \gamma(-\lambda_i) M_i p_j + \gamma(-\lambda_j) M_j p_i - \gamma(\lambda_i) M_i p_{ij} - \gamma(\lambda_j) M_j p_{ij}, \quad (14)$$

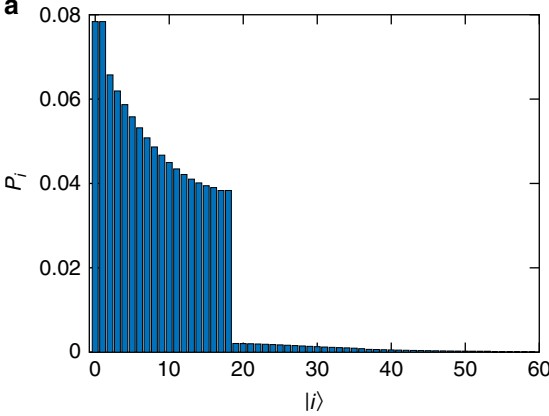

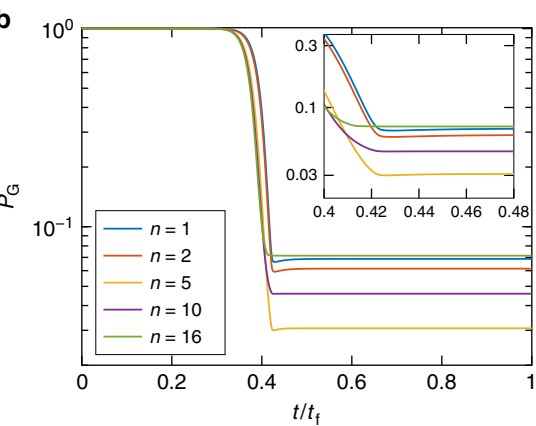

**Fig. 4** Master equation results for the state populations when restricting the excited states to single-fermion states. **a** The population in each single-fermion state at $t = t_f$ in a one-fermion simulation. The chain parameters are $N = 176$, $W_1 = 1$, $W_2 = 0.5$, $t_f = 5\,\mu s$, and $n = 5$. With the annealing schedule given in Methods, the quantum minimum gap is at $s^* = t^*/t_f \approx 0.424$. At this point we find $k^* = 18$ single-fermion states below the thermal energy $T = 12$ mK (D-Wave processor operating temperature). As expected, in one-fermion simulations, most of the population is found in the first $k^*$ states. A long tail of more energetic single-particle states beyond the first $k^*$ retain some population. **b** Evolution of the instantaneous ground state populations for ASCs with the same parameters as in (**a**), but for different sector sizes $n$ and with two-fermion states. The ground state loses the majority of its population as it approaches the minimum gap point at $t/t_f = s^*$. The largest drop is found for $n = n^* = 5$. Inset: magnification of the region around the minimum gap. Relaxation plays essentially no role. Instead, the population freezes almost immediately

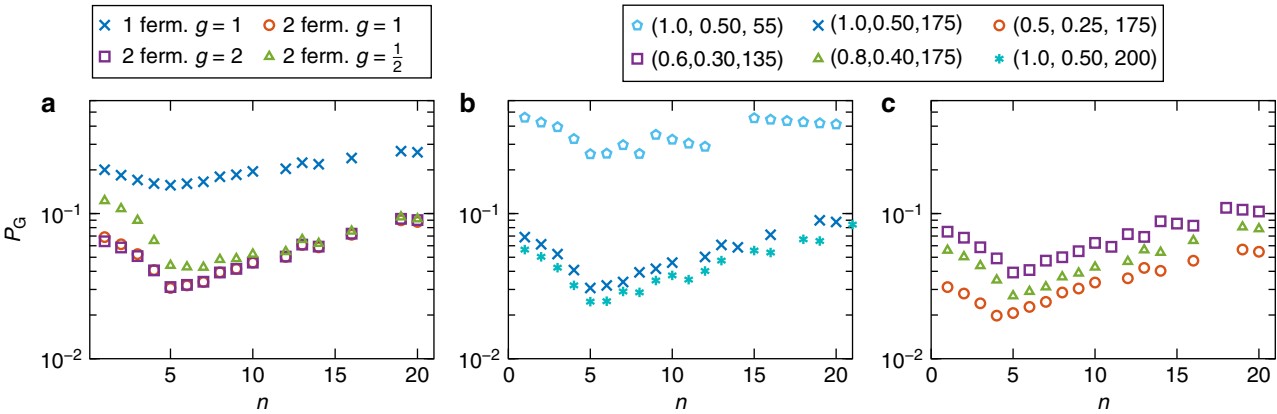

**Fig. 5** Master equation results for the ground state population when restricting the excited states to single and two-fermion states. **a** The result of simulating the ASC problem with parameters (1,0.5,175) via the adiabatic Pauli master equation (8), restricted to the vacuum + single-fermion states, and vacuum + single-fermion + two-fermion states. Also shown is the dependence on the system-bath coupling parameter $g$ in the two-fermion case; doubling it has little impact, whereas halving it increases the success probability somewhat for $n < 14$. The position of the minimum at $n^* = 5$ matches the empirical result seen in Fig. 2a, except when $g = 1/2$, i.e., the position is robust to doubling $g$ but not to halving it. Panels (**b**) and (**c**) show additional 2-fermion master equation results with $g = 1$. Note that for the (1,0.5,200) chain, these simulations exhibit better agreement with the DW2X data than the simple $k^*$ analysis plotted in Fig. 2d–f. This is because the simulations also keep track of the Boltzmann factor

where all summations run from 1 to $k^*$, and $p_{ij}$ denotes the population in the two-particle fermion energy state $\eta_i^\dagger \eta_j^\dagger |0\rangle$.

We can now numerically solve this system of equations. As seen in Fig. 4a, where we plot the final populations in the different single-particle fermion states at $t = t_f$ for one-fermion simulations, only the first $k^*$ single-fermion levels are appreciably populated. This agrees with our aforementioned assumption that states with energy greater than $\lambda_{k^*}$ are not thermally populated. In Fig. 4b we plot the population in the instantaneous ground state as a function of time for two-fermion simulations. The system starts in the gapped phase where the ground state population is at its chosen initial value of 1. The ground state rapidly loses population via thermal excitation as the system approaches the critical point, after which the population essentially freezes, with repopulation via relaxation from the excited states essentially absent (see inset). Thus, it is not relaxation that explains the increase in ground state population seen in Fig. 2a–c for $n > n^*$. Instead, we find that the ground state population drops most deeply for $n = n^*$. This, in turn, is explained by the behavior of $k^*$ seen in Fig. 2d–f, as discussed earlier.

We show in Fig. 5 the predicted final ground state population under the one and two-fermion restriction. This minimal model already reproduces the correct location of the minimum in $P_G$. It also reproduces the nonmonotonic behavior of the success probability. It does not correctly reproduce the exponential fall and rise. However, including the two-fermion states gives the right trend: it leads to a faster decrease and increase in the population without changing the position of the minimum, suggesting that a simulation with the full $2^{k^*}$ states would recover the empirically observed exponential dependence of the ground state population seen in Fig. 2a–c.

**Discussion**
A commonly cited failure mode of closed-system quantum annealing is the exponential closing of the quantum gap with increasing problem size. It is expected, on the basis of the Landau–Zener formula and the quantum adiabatic theorem, that to keep the success probability of the algorithm constant the run-time should increase exponentially. As a consequence, one expects the success probability to degrade at constant run-time if the gap decreases with increasing problem size. Our goal in this work was to test this failure mode in an open-system setting

**Table 1 Chain length (N) and sector size (n) for N ~ 175**

| $N$ | 174 | 175 | 172 | 173 | 176 | 175 | 176 | 169 |
|---|---|---|---|---|---|---|---|---|
| $n$ | 1 | 2 | 3 | 4 | 5 | 6 | 7 | 8 |
| $N$ | 172 | 171 | 181 | 170 | 183 | 177 | 172 | 181 |
| $n$ | 9 | 10 | 12 | 13 | 14 | 16 | 19 | 20 |

where the temperature energy scale is always larger than the minimum gap. We did so by studying the example of a ferromagnetic Ising chain with alternating coupling-strength sectors, whose gap is exponentially small in the sector size, on a quantum annealing device. Our tests showed that while the success probability initially drops exponentially with the sector size, it recovers for larger sector sizes. We found that this deviation from the expected closed-system behavior is qualitatively and semi-quantitatively explained by the system's spectrum around the quantum critical point. Specifically, the scaling of the quantum gap alone does not account for the behavior of the system, and the scaling of the number of energy eigenstates accessible via thermal excitations at the critical point (the thermal density of states) explains the empirically observed ground state population.

Does there exist a classical explanation for our empirical results? We checked and found that the SVMC model[31] is capable of matching the empirical DW2X results provided we fine-tune its parameters for each specific chain parameter set $\{W_1, W_2, N\}$. However, it does not provide as satisfactory a physical explanation of the empirical results as the fermionic or master equation models, which require no such fine-tuning; see Supplementary Note 6 for details.

Our work demonstrates that care must be exercised when inferring the behavior of open-system quantum annealing from a closed-system analysis of the scaling of the gap. It has already been pointed out that quantum relaxation can play a beneficial role[26–30]. However, we have shown that relaxation plays no role in the recovery of the ground state population in our case. Instead, our work highlights the importance of a different mechanism: the scaling of the number of thermally accessible excited states. Thus, to fully assess the prospects of open-system

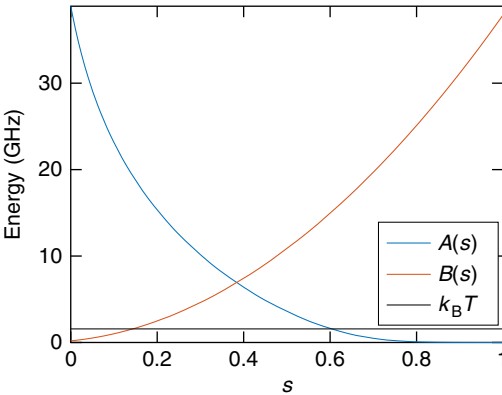

**Fig. 6** Annealing schedules and temperature $A(s)$ and $B(s)$ are the annealing schedules of the D-Wave 2X processor used in this work. The fridge temperature (horizontal black line) is $T = 12$ mK

quantum annealing, this mechanism must be understood along with the scaling of the gap and the rate of thermal relaxation. Of course, ultimately we only expect open-system quantum annealing to be scalable via the introduction of error correction methods[51,57–59].

## Methods

**Alternating sector chains**. We generated a set of ASCs with chains lengths centered at $N \sim \{55, 135, 175, 200\}$ and with sector sizes $n$ ranging from 2 to 20. Since the chain length and sector size must obey the relation $(N - 1)/n = 2b + 1$ with integer $b$, there is some variability in $N$. Table 1 gives the $(N, n)$ pair combinations we used for chain set with mean length 175.

**Quantum annealing processor used in this work**. The D-Wave 2X processor (DW2X) is an 1152-qubit quantum annealing device made by D-Wave Systems, Inc., using superconducting flux qubits[52]. The particular processor used in this study is located at the University of Southern California's Information Sciences Institute, with 1098 functional qubits and an operating temperature of 12 mK. The total annealing time $t_f$ can be set in the range [5, 2000] μs. The time-dependent Hamiltonian the processor is designed to implement is given by

$$H(s) = A(s) \sum_i \sigma_i^z + B(s) \left( \sum_i h_i \sigma_i^z + \sum_{(i,j)} J_{ij} \sigma_i^z \sigma_j^z \right), \quad (15)$$

with dimensionless time $s = t/t_f$. Figure 6 describes the annealing schedules $A(s)$ and $B(s)$. The coupling strengths $J_{ij}$ between qubits $i$ and $j$ can be set in the range $[-1, 1]$ and the local fields $h_i$ can be set in the range $[-2, 2]$.

We used $t_f = 5$ μs. For each ASC instance we implemented 10 different embeddings, with 10 gauge transforms each[53]. In total, $10^5$ runs and readouts were taken per instance. The reported success probability is defined as the fraction of readouts corresponding to a correct ground state. For additional details on the DW2X processor we used see, e.g., Ref.[54].

**Numerical procedure for solving the master equation**. We solve the coupled differential Eqs. (11)–(14) using a fourth order Runge–Kutta method given by Dormand and Prince[55] with nonnegativity constraints[56]. We compute the transition matrix elements via Supplementary Eq. (35) and the bath correlation term via Eq. (10).

**Data availability**. The data that support the findings of this study are available from the corresponding author upon reasonable request.

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

## Acknowledgments

We thank Ben W. Reichardt for useful discussions. The computing resources used for this work were provided by the USC Center for High Performance Computing and Communications. A.M. was supported by the USC Provost Ph.D. Fellowship. The research is based upon work (partially) supported by the Office of the Director of National Intelligence (ODNI), Intelligence Advanced Research Projects Activity (IARPA), via the U.S. Army Research Office contract W911NF-17-C-0050. The views and conclusions contained herein are those of the authors and should not be interpreted as necessarily representing the official policies or endorsements, either expressed or implied, of the ODNI, IARPA, or the U.S. Government. The U.S. Government is authorized to reproduce and distribute reprints for Governmental purposes notwithstanding any copyright annotation thereon.

## Author contributions

A.M., T.A., and D.A.L. designed the experiments, which A.M. performed. A.M., T.A. and D.A.L. contributed to the analysis and the writing of the manuscript. A.M.computed the thermal transition probabilities and performed the numerical simulations.

## Additional information

**Competing interests:** The authors declare no competing interests.

