## [Peer Review File · Nature Communications]

Reviewers' comments:

Reviewer #1 (Remarks to the Author):

The performance of quantum annealers at finite temperature is a question of both basic scientific and technological importance. Because the gap between adjacent energy levels in the spectrum will be smaller than the thermal energy kT in the limit of large problem size, thermal effects will generically be important. Is there a regime in which quantum annealing can remain quantum?

This question has been the subject of considerable prior investigation - [28] was the first (2008) serious consideration of whether thermal effects can improve the performance of quantum annealers. The theoretical predictions of [29] were supported by the experiments in [29].

The present manuscript adds to our understanding of this question by providing a detailed analysis of an exactly solvable case of the transverse ising model, which nevertheless suffers from exponentially small energy gaps. The authors show that the observed behavior of the success probability with system size can be reproduced qualitatively, and in some aspects quantitatively, by an open systems model in which the number of thermally accessible excited states is the key variable.

While the results presented here are novel and provide new experimental and theoretical results that contribute to our understanding of quantum annealing at finite temperature, their contextualization in the literature is significantly inaccurate in ways that reduces the impact of the reported work.

The authors refer in particular to [28] and [29] as papers that rely only on thermal relaxation. This is not correct. In [28] the evolution is divided into three regions, and only in the last region is the dominant effect thermal relaxation. In fact [28] introduces the idea that significant mixing due to open system effects (beyond relaxation) at an anti crossing between the first excited and ground states could provide an advantage. This also implies that the higher lying levels (above the first excited state) are potential sources of error for such thermally assisted quantum annealing.

This is the point that is made so strongly by the current paper - the authors explicitly observe and quantify these sources of error in a model where both exact analysis and experimental implementation are possible. In [28] exact analysis of adiabatic grover search was performed, a

model which cannot be experimentally implemented in a transverse field Ising model, and numerical simulations of random field Ising models.

Proper contextualization of these results in the prior literature will strengthen the authors points and improve the impact of the results in the paper. If these changes can be made I strongly recommend publication in Nature Communications

Reviewer #2 (Remarks to the Author):

This is a very interesting, and potentially provocative, paper.

The authors utilize a D-Wave 2000-qubit machine to study a one-dimensional (1d) linear chain system. The chain system is the ferromagnetic transverse Ising spin chain, with open boundary conditions, and 'alternating sector chain' (ASC) with sector length n . The authors find the probability the D-Wave returns the correct ground state for the ferromagnetic chain has a minimum in n .

This observation goes against the predictions for quantum annealers (QA) in closed quantum systems, where success should scale only with the energy gap. The authors make a good case their observations are due first to the nature of an open quantum system at finite temperature and second due to the number of single-fermion energies less than the energy associated with the device temperature. They justify their additional assumption in the D-Wave, and for their parameters, only single-fermion transitions are involved. In other words, only a single fermion is changed at any time.

Comments the authors may wish to address:

*)A closed quantum system analysis predicts scaling depends only on the energy gap, while the analysis here for an open quantum system for their D-Wave data depends only on the number of free fermion energies below the energy of the device temperature. In general, do the authors expect both effects to be present? If so, can they manipulate their parameters to test on the D-Wave a region where both effects are important?

*)The major problem with this paper is the 1d Ising ferromagnet is in the complexity class P. The D-Wave and other QA are designed to provide solutions to problems which are known to be in other sectors of NP (assuming NP is not equal to P). How do the authors expect their results will translate to the more typical case where the problem has frustration?

Comments for readability:

*)I suggest in the Supplementary Information the authors put in a graphic to more quickly explain the ASC, as well as the n , N , b , W_1 (heavy), and W_2 (light). This is particularly true since the authors do not define their notation in Eq.3.

*)The authors never adequately define the quantum critical point. This is central to their analysis. For the non-expert, it may not be clear what time during the anneal they are talking about, and where that falls in the D-Wave annealing cycle.

*)Due to the finite chain size N , the authors should also state whether they see any effects of finite size scaling with N for fixed n . The prediction would be that small N should make the effect they see less pronounced, because the critical point only exists in the limit of infinite N .

*)The authors use the 5 microsecond default on the D-Wave as the annealing time. They should state somewhere the effect of changing to 2000 microsec. As seen, for example in Supplementary Figure 2, this probably will not significantly change the results.

*)In the Supplementary section on the SVMC model, it is unclear whether the fitting they show is a typical result, best result, or worst result of attempted fits to their D-Wave data.

Reviewer #3 (Remarks to the Author):

The authors investigate quantum annealing in an Alternating Sectors Chain (ACS) model, a spin chain with alternating sectors of strong and weak ferromagnetic spin-spin coupling. In this model quantum annealing is expected to encounter an exponentially small gap. The size of the sector controls the spectrum of the model and the minimal gap in particular, and is used as a tunable parameter. The observed dependence of the ground state probability at the end of the annealing protocol on the sector size is non-monotonic and cannot be explained by the dependence of the minimal gap on the sector size. The authors show that this behavior is consistent with open system dynamics of quantum Ising chain coupled to a thermal bath. Specifically, the authors use Jordan-Wigner mapping onto a fermionic model to estimate the rate of escape from the ground state which is determined by the number of thermally accessible states at the critical point in the course of the annealing protocol. The authors find that the dynamics at the critical point determines the final ground state probability due to the rapid freezing of the dynamics with decreasing transverse field beyond the critical point. This estimate explains the experimentally observed dependence of the ground state population on the sector size. Authors also attempt to fit the experimental data using spin vector Monte Carlo algorithm, which corresponds to strong dissipation (classical) limit of the spin chain dynamics, however only partial consistency is found suggesting that a quantum behavior is likely observed in the experiment. Alternative classical explanations cannot be completely ruled out nonetheless the authors make a substantive case for interpretation of the data in terms of quantum fermion model at a critical point.

The results in the paper are novel and will attract very wide interest. Particularly the author's finding that quantum open system dynamics at the critical point strongly affects the success of quantum annealing will be of interest for the growing community focused on quantum annealing. The experimental observation of the effects of quantum critical dynamics in quantum annealer would be of interest to a wider physics community.

However I have several comments to this study.

I would recommend the paper publication in the Nature Communications Journal provided that authors address the comment provided below.

General comments:

I wanted to mention the previous study of the effect of environment on quantum annealing in linear transverse field Ising chain with uniform Ising coupling (Quantum Annealing via Environment-Mediated Quantum Diffusion, Phys. Rev. Lett. 118, 066802). Despite the difference in models, the results of that paper are relevant to the present study. Specifically, these results describe what happens in the region after the critical point. In that region the environment causes quantum diffusion that gives rise to significant deviations from the rate equation-based dynamics. This happens because an elementary act of fermion recombination requires 2 fermions to approach each other, and at low concentration this process becomes fermion diffusion-limited. It takes a longer time for fermions to reach each other than the act of recombination itself. This affects the

final fermion density and quantum annealing performance. As mentioned above the analysis of this diffusion required going beyond the rate equation approximation used in the present paper because the fermion momentum relaxation becomes important. In the cited paper it was done using Green function methods. I think this result can be important in all fermionizable time-dependent quantum spin chain models coupled to environment where a large number of many fermions are generated at the criticality. This is the case in the present study. I think it is important that authors refer to the aforementioned paper and mention the role of the diffusion-limited quantum relaxation at low dimensional systems.

Comments on the text:

- (i) 0; Fig1 (a), Inset: the units on the horizontal axis of the inset are not fully clear. A clarification in the caption or showing units explicitly as inset of Fig 1d would help.
- (ii)  A0; Fig S1 (supplementary material) is important for theoretical interpretation of the data and needs to be in the main text.
- (iii)  OAO; Prior theoretical analysis discussed thermal population dynamics in the course of quantum annealing in an Ising chain and needs to be acknowledged Phys. Rev. Lett. 118, 066802 (2017)
- (iv)  A0;Page 4 paragraph 1 figure labels appear mixed up after words: "...k^{*} peaks at n^{*}=5..."

Dear Editors and Referees,

We thank the Referees for carefully reading our paper. We appreciate the positive and constructive critique of our manuscript. We respond to the comments below.

Referee 1:

The authors refer in particular to Amin et al. [1] and Dickson et al. [2] as papers that rely only on thermal relaxation. This is not correct. In [1] the evolution is divided into three regions, and only in the last region is the dominant effect thermal relaxation. In fact [1] introduces the idea that significant mixing due to open system effects (beyond relaxation) at an anti crossing between the first excited and ground states could provide an advantage. This also implies that the higher lying levels (above the first excited state) are potential sources of error for such thermally assisted quantum annealing.

This is the point that is made so strongly by the current paper - the authors explicitly observe and quantify these sources of error in a model where both exact analysis and experimental implementation are possible. In [1] exact analysis of adiabatic grover search was performed, a model which cannot be experimentally implemented in a transverse field Ising model, and numerical simulations of random field Ising models.

Proper contextualization of these results in the prior literature will strengthen the authors points and improve the impact of the results in the paper. If these changes can be made I strongly recommend publication in Nature Communications.

We thank the Referee for bringing the issue to our attention. We did not mean to imply that the cited works relied solely on thermal relaxation to improve the success probability; as the Referee correctly points out, thermal-assistance in QA can take other forms. We have adjusted the wording to make it clear that we are only stating that thermal relaxation is not the dominant effect in our results. In addition we significantly expanded an introductory paragraph in order to properly contextualize Amin *et al.* [1] and Dickson *et al.* [2].

Referee 2:

We give a point-by-point response to the Referee’s comments below.

1. *A closed quantum system analysis predicts scaling depends only on the energy gap, while the analysis here for an open quantum system for their D-Wave data depends only on the number of free fermion energies below the energy of the device temperature. In general, do the authors expect both effects to be present? If so, can they manipulate their parameters to test on the D-Wave a region where both effects are important?*

These are important questions. Briefly, the answers are ‘yes’ to the first and ‘no’ to the second. In more detail:

(1) As we note in the last paragraph under section “Spectral analysis” and Supplementary Note B, both the gap and 2^{k^*} (recall that k^* is the number of single-fermion states with energy smaller than the thermal gap at the critical point and that $\sim 2^{k^*}$ states are thermally accessible from the ground state) play a role in determining the success probabilities of the ASCs in the open-quantum system setting.

(2) Due to the structure of the ASCs’ spectrum and the strength of coupling to the thermal environment on the D-Wave device, the change in the value of k^* dominates over the change in the value of gap Δ . Ideally, if we could reduce the coupling strength to the environment or the temperature, we would expect that the closed system intuition based on the minimum gap to become more and more relevant. Unfortunately, we do not have this capability on the D-Wave devices.

2. *The major problem with this paper is the 1d Ising ferromagnet is in the complexity class P. The D-Wave and other QA are designed to provide solutions to problems which are known to be in other sectors of NP (assuming NP is not equal to P). How do the authors expect their results will translate to the more*

typical case where the problem has frustration?

The problem hardness for QA is typically quantified in terms of the minimum gap scaling (see Ref. [3] for collected examples). This quantification relies on the closed-system adiabatic condition, which is the relevant scenario strictly when we have access to an ideal device with no decoherence. Our work shows that the hardness and hence performance for a physical device may be more complicated when open-system effects are taken into account. Both the minimum gap and the number of excited states may determine the success of the device in solving the problem. Therefore, our work suggests that when searching for problems that might be amenable to be solved on a QA device in which open-system effects are non-negligible, having both a large minimum gap and a small number of low-lying states are likely critical for success. This conclusion is likely to be applicable beyond the specific ASC problem we have considered here, and we expect it to apply in particular also to problems with frustration.

3. *I suggest in the Supplementary Information the authors put in a graphic to more quickly explain the ASC, as well as the n , N , b , W_1 (heavy), and W_2 (light). This is particularly true since the authors do not define their notation in Eq. (3)*

Thank you for this suggestion. The new Figure 1 now shows a sample chain with $n = 3$, $N = 10$ and $b = 1$.

4. *The authors never adequately define the quantum critical point. This is central to their analysis. For the non-expert, it may not be clear what time during the anneal they are talking about, and where that falls in the D-Wave annealing cycle.*

Thank you also for this suggestion. We have amended the final paragraph under the Fermionization section. The added text defines the quantum critical point for this problem and its location in the annealing cycle:

“In the thermodynamic limit, this point coincides with the quantum critical point where the geometric mean of the Ising fields balances the transverse field, $A(s^*) = \sqrt{W_1 W_2} B(s^*)$ [4, 5].”

5. *Due to the finite chain size N , the authors should also state whether they see any effects of finite size scaling with N for fixed n . The prediction would be that small N should make the effect they see less pronounced, because the critical point only exists in the limit of infinite N .*

The Referee is correct that the effects are less pronounced for smaller chain sizes. For sufficiently low temperature and small chains, we expect the effect to disappear entirely. In Figs. 1(b), 1(e) and 3(b), we discussed three chains with a similar range of n values but with three different values of $N \in \{55, 175, 200\}$. We notice that the minima becomes more pronounced as we increase our chain size. For fixed n and increasing N , we observe an increase in k^* and decrease in Δ , but the location of the maxima of k^* is fixed. We have highlighted this result in the caption of Fig. 2 with the text:

“The value of n^* (minimum of P_G) is observed to be independent of the chain length N for the sizes studied. Longer chains result in a lower P_G and a more pronounced minimum.(...) Longer chains result in a larger value of k^* and a more pronounced maximum.”

6. *The authors use the 5 microsecond default on the D-Wave as the annealing time. They should state somewhere the effect of changing to 2000 microsecond. As seen, for example in Supplementary Figure 2, this probably will not significantly change the results.*

We thank the Referee for this suggestion. As the Referee noted, we do not expect that increasing the annealing time will substantially change the qualitative behavior of our results. Since the magnitude of thermal excitation and relaxation depends on the time spent by the system near the critical point, we do expect a quantitative change in the success probabilities. In order to collect sufficient data, we restricted our maximum annealing time to $t_f = 1000 \mu s$. In Fig. 1 of this reply, we show additional data for the ASC with parameters (1.0, 0.5, 175) and with $t_f \in \{5, 200, 500, 1000\} \mu s$. The effect is to change the success probabilities, but not the location of the minimum. We have added this new data to an additional supplementary section E and referenced it in the results section.

Figure 1: Dependence of the success probability on the annealing times for the ASC with parameters (1.0, 0.5, 175). The location of the minimum is unchanged as the annealing time is varied.

7. *In the Supplementary section on the SVMC model, it is unclear whether the fitting they show is a typical result, best result, or worst result of attempted fits to their D-Wave data.*

We have shown the best fit that we found for the ASC with parameters (1.0, 0.5, 175). In general, SMVC can exhibit a wide range of behavior. To clarify this we have appended the following text to the fourth paragraph of Supplementary Note D:

“This is the best fit we found for this particular chain. In general, we found that the SVMC parameters can be tuned to reproduce the location of the minimum in success probability for any sector size. We were also able to tune the parameters such that the minimum disappears completely and have the success probability increase or decrease monotonically. We were not able to find parameters that give rise to an inverted curve, i.e., a maximum in the success probability. Most of these features can be seen by tuning β and keeping the other parameters fixed.”

Referee 3:

I wanted to mention the previous study of the effect of environment on quantum annealing in linear transverse field Ising chain with uniform Ising coupling [6]. Despite the difference in models, the results of that paper are relevant to the present study. Specifically, these results describe what happens in the region after the critical point. In that region the environment causes quantum diffusion that gives rise to significant deviations from the rate equation-based dynamics. This happens because an elementary act of fermion recombination requires 2 fermions to approach each other, and at low concentration this process becomes fermion diffusion-limited. It takes a longer time for fermions to reach each other than the act of recombination itself. This affects the final fermion density and quantum annealing performance. As mentioned above the analysis of this diffusion required going beyond the rate equation approximation used in the present paper because the fermion momentum relaxation becomes important. In the cited paper it was done using Green function methods. I think this result can be important in all fermionizable time-dependent quantum spin chain models coupled to environment where a large number of many fermions are generated at the criticality. This is the case in the present study. I think it is important that authors refer to the aforementioned paper and mention the role of the diffusion-limited quantum relaxation at low dimensional systems.

We thanks the Referee for bringing this important work to our notice. We have added this relevant observation in the text as follows: “Longer annealing times do not change the qualitative behavior of the results,

but do lead to changes in the success probability (we provide these results in Supplementary note E). A longer annealing time can result in more thermal excitations near the minimum gap, but it may also allow more time for ground state repopulation after the minimum gap. The latter can be characterized in terms of a recombination of fermionic excitations by a quantum-diffusion mediated process [6]. Unfortunately, we cannot distinguish between these two effects, as we only have access to their combined effect in the final-time success probability.”

We give a point-by-point response to the rest of the Referee’s comments below.

- i) *Fig. 1(a), Inset: the units on the horizontal axis of the inset are not fully clear. A clarification in the caption or showing units explicitly as inset of Fig 1d would help.*

To maintain readability of the entire figure set, we are unable to add a description to the axis of the inset itself. In order to avoid confusion, we have added additional details in the caption of Fig. 2 (the old Fig.1) which now reads:

“The minimum gap (in GHz) of the chains (calculated using the Jordan-Wigner transformation) as a function of the sector size $n \in \{1, \dots, 20\}$ ”.

- ii) *Fig S1 (supplementary material) is important for theoretical interpretation of the data and needs to be in the main text.*

We have moved the figure to the main text.

- iii) *Page 4 paragraph 1 figure labels appear mixed up after words: “ k^* peaks at $n^* = 5$ ”*

We thank the Referee for bringing the typo to our attention. We have fixed the label accordingly. We have changed

“ k^* peaks at $n^* = 5$ [Fig. 1(b)] but the empirical success probability for $n = 5$ and $n = 6$ is roughly the same [Fig. 1(e)].”

to

“where k^* peaks at $n^* = 5$ [Fig. 1(e)] but the empirical success probability for $n = 5$ and $n = 6$ is roughly the same [Fig. 1(b)].”

References

- [1] M. H. S. Amin, P. J. Love, and C. J. S. Truncik, “Thermally assisted adiabatic quantum computation,” *Phys. Rev. Lett.* **100**, 060503 (2008).
- [2] N. G. Dickson, M. W. Johnson, M. H. Amin, R. Harris, F. Altomare, A. J. Berkley, P. Bunyk, J. Cai, E. M. Chapple, P. Chavez, F. Cioata, T. Cirip, P. deBuen, M. Drew-Brook, C. Enderud, S. Gildert, F. Hamze, J. P. Hilton, E. Hoskinson, K. Karimi, E. Ladizinsky, N. Ladizinsky, T. Lanting, T. Mahon, R. Neufeld, T. Oh, I. Perminov, C. Petroff, A. Przybysz, C. Rich, P. Spear, A. Tcaciuc, M. C. Thom, E. Tolkacheva, S. Uchaikin, J. Wang, A. B. Wilson, Z. Merali, and G. Rose, “Thermally assisted quantum annealing of a 16-qubit problem,” *Nat. Commun.* **4**, 1903 (2013).
- [3] T. Albash and D. A. Lidar, “Adiabatic quantum computation,” *Reviews of Modern Physics* **90**, 015002–(2018).
- [4] J. Hermisson, U. Grimm, and M. Baake, “Aperiodic ising quantum chains,” *Journal of Physics A: Mathematical and General* **30**, 73157335 (1997).
- [5] P. Pfeuty, “An exact result for the 1d random ising model in a transverse field,” *Physics Letters A* **72**, 245246 (1979).
- [6] V. N. Smelyanskiy, D. Venturelli, A. Perdomo-Ortiz, S. Knysh, and M. I. Dykman, “Quantum annealing via environment-mediated quantum diffusion,” *Physical Review Letters* **118** (2017), 10.1103/physrevlett.118.066802.

REVIEWERS' COMMENTS:

Reviewer #1 (Remarks to the Author):

The authors have addressed my comments. I recommend this paper for publication.

Reviewer #2 (Remarks to the Author):

I have re-read this interesting manuscript, as well as analyzed the replies and modifications of the authors due to my review comments and those of the other two reviewers.

I now believe this article has been improved, and should be published in Nature Communications.

Reviewer #3 (Remarks to the Author):

I am satisfied with the changes and believe that the manuscript should be accepted to the Nature Communications in the present form.

Dear Editors and Referees,

We appreciate the response to our revised manuscript. We thank all the three referees for their positive critique and the quick review.

Referee 1:

The authors have addressed my comments. I recommend this paper for publication.

Referee 2:

I have re-read this interesting manuscript, as well as analyzed the replies and modifications of the authors due to my review comments and those of the other two reviewers.

I now believe this article has been improved, and should be published in Nature Communications.

Referee 3:

I am satisfied with the changes and believe that the manuscript should be accepted to the Nature Communications in the present form.